# Rift Valley fever virus dynamics in a transhumant cattle system in The Gambia

Essa Jarra[1,2]*, Divine Ekwem[2], Sarah Cleaveland[2], Daniel T Haydon[2]*

[1]Department of Livestock Services, Ministry of Agriculture, Abuko, The Gambia; [2]School of Biodiversity, One Health, and Veterinary Medicine, College of Medical, Veterinary and Life Sciences, University of Glasgow, Glasgow, United Kingdom

## eLife Assessment

This modelling study tests several hypotheses describing how seasonality and migration drive the epidemiology of Rift Valley Fever Virus among transhumant cattle in The Gambia. The work is methodologically **solid**, and the findings offer **valuable** insights into how the movement of cattle in and out of the Gambia River and Sahel ecoregions could lead to source-sink transmission dynamics among cattle subpopulations, sustaining endemic transmission.

*For correspondence:
jarraessa@yahoo.com (EJ);
Daniel.Haydon@glasgow.ac.uk (DTH)

Competing interest: The authors declare that no competing interests exist.

**Abstract** Rift Valley fever (RVF) is a zoonotic disease of global concern, driven by environmental conditions, vector activity, and livestock mobility. Although RVF has been reported in The Gambia, its epidemiology remains poorly understood. This study developed a compartmental model to study RVF dynamics in the cattle population of the country. The model incorporated seasonally dynamic transmission parameters reflecting transhumant movement and ecological differences between two distinct ecoclimatic regions: the Sahelian area and the Gambia river. Parameterised using serological data linked to household survey data, the model predicted endemic RVF virus (RVFV) circulation within The Gambia and captured temporal infection trends that closely match empirical data. Weak decay rates of seropositivity were required to match predicted and observed age-seroprevalence. Results indicated sustained RVFV transmission during the dry season in the Gambia river eco-region, with a high risk of seasonal virus introductions to the Sahelian eco-region at the start of the wet season via the returning transhumant cattle. Our study highlighted the role of livestock mobility in RVFV epidemiology in The Gambia and the need for targeted control strategies that might include, for example, targeted cattle vaccination or application of topical insecticide treatments for transhumant herds.

## Introduction

Livestock production is a cornerstone of rural livelihoods across sub-Saharan Africa, contributing to economic stability, socio-cultural cohesion, and food security (*Herrero et al., 2013*). However, the sector is increasingly threatened by infectious diseases and climate change impacts – ranging from conditions intensifying vector proliferation to those exacerbating water and pasture scarcity (*Stanimirova et al., 2019*; *Thornton et al., 2009*). In this context, transhumant and nomadic practices across diverse settings serve as critical resilience strategies (*Aryal et al., 2018*; *Motta et al., 2018*), allowing pastoral communities to respond to seasonal resource challenges. In the West African Sahel, transhumant movements typically lead to resource-rich regions along major rivers where abundance of water and high-quality pasture support large populations of livestock (*Bassett and Turner, 2007*; *Freudenberger and Freudenberger, 1993*). Despite their importance, these movements are poorly understood due to their largely unregulated nature and limited documentation (*Belkhiria et al., 2019*;

*Erdaw, 2023*). The association between livestock movements and the spread of infectious diseases has long been recognised (*Fèvre et al., 2006*; *Kim et al., 2021*; *Prentice et al., 2017*), allowing introduction of pathogens to local susceptible populations. The spread of infectious diseases is further exacerbated by the very limited veterinary surveillance and disease prevention services available to nomadic and transhumant pastoralists (*Belkhiria et al., 2019*; *Schelling et al., 2008*), who often inhabit remote and hard-to-reach regions.

The epidemiology of Rift Valley fever (RVF), a WHO priority zoonotic disease caused by the Rift Valley fever virus (RVFV) (family: Phenuiviridae), exemplifies the interplay between ecological conditions, vector activity, and livestock mobility (*Durand et al., 2020*; *Walsh et al., 2017*). RVFV is primarily transmitted by mosquitoes, with competent species identified across the *Aedes*, *Culex*, and *Anopheles* genera (*Linthicum et al., 2016*; *Lumley et al., 2017*). In addition to vector transmission, both human and ruminant infections can also occur through direct contact with infectious tissues and fluids such as blood and milk during outbreaks (*Gerken et al., 2022*). Outbreaks are characterised by abortions and neonatal mortalities in sheep and goats (*Ksiazek et al., 1989*; *Munyua et al., 2010*). Although cattle and camels are generally less susceptible to RVFV infection compared to sheep (*Nair et al., 2023*), the movements of asymptomatic but infectious cattle within and between districts (particularly along cattle corridors linking grazing and water resources) can transport the virus from areas of active transmission to where no outbreaks have been reported (*Bird et al., 2009*; *Shoemaker et al., 2002*; *Tumusiime et al., 2023*). Human cases range from asymptomatic or mild flu-like symptoms to a severe and sometimes fatal syndrome, with complications including haemorrhagic fever or encephalitis (*Hoogstraal et al., 1979*; *Madani et al., 2003*). Since its discovery in Kenya in 1931 (*Daubney et al., 1931*), RVF has caused repeated outbreaks across sub-Saharan Africa, with notable expansions into Egypt (*Hoogstraal et al., 1979*), the Arabian Peninsula (*Balkhy and Memish, 2003*), and Comoros Archipelago (*Sissoko et al., 2009*), underscoring its global threat potential.

In The Gambia, situated in the Sahelian zone of West Africa, the livestock sector contributes approximately 30% of the agricultural GDP and 10% of the national GDP (*Rich et al., 2020*). The majority of livestock management practices are traditional, with 98% of livestock-owning households engaged in either pastoral (entirely dependent on livestock production) or semi-pastoral systems (integrating crop husbandry and livestock rearing) (*Department of Livestock Services, 2016*, pers. comms.). The floodplains of the Gambia river serve as a perennial resource hub, providing water and pasture that attract transhumant herds migrating from the Sudano-Sahelian zone during the prolonged harsh dry season (November to June). Additionally, riverine ecosystems offer ideal breeding sites for RVFV-competent mosquito species in the *Culex* and *Mansonia* genera (*Diallo et al., 2000*; *Saluzzo et al., 1984*; *Snow, 1983*), supporting year-round mosquito proliferation. The Gambia river floodplains thus present an environment conducive to the spread of both vector-borne and non-vector-borne livestock diseases. RVF cases affecting both humans and livestock were first reported in The Gambia in 2002 (*Food and Agriculture Organisation, 2003*); however, surveillance remains limited, leaving critical gaps in understanding of its epidemiology. Our recent serological investigation revealed ongoing RVFV transmission within ruminant livestock and highlighted the importance of transhumant movement to the Gambia river as a risk factor for livestock seropositivity (*Jarra et al., 2025*). These findings underscore the need to explore how livestock movement and ecological variability interact to shape RVF dynamics in The Gambia.

In a disease system where transmission is influenced by the movement of hosts, the incorporation of movement processes between habitats is essential for accurately modelling its dynamics (*Merkle et al., 2018*). Recent studies have integrated livestock movement patterns into the classical compartmental model framework to enhance our understanding of RVF dynamics. For instance, simulations have explored how nomadic ruminants, which occasionally or seasonally migrate from pond to pond according to the availability of water, sustain endemic RVFV circulation in the Ferlo region of Senegal (*Durand et al., 2020*). Similarly, an earlier study demonstrated that localised environmental conditions and inter-island livestock movements could sustain RVFV circulation in the Comoros Archipelago, without reintroduction from mainland Africa. Notably, this study used vegetation index as a proxy parameter for mosquito-borne transmission in a data-constraint context and achieved predictions consistent with empirical observations (*Tennant et al., 2021*). While these studies provide valuable insights, their focus on specific ecological and epidemiological contexts limits their applicability to The Gambia, which presents distinct ecological and livestock movement characteristics.

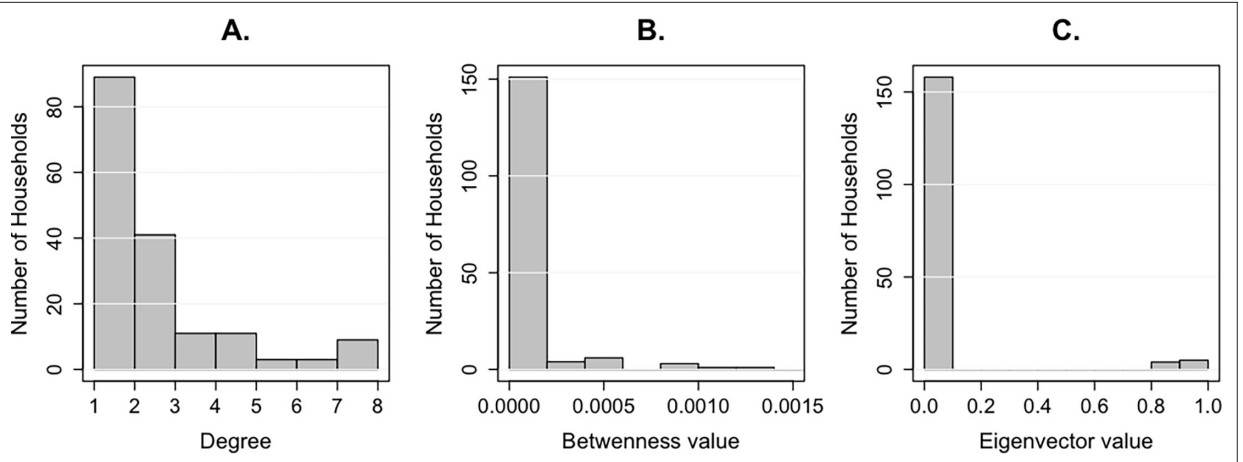

**Figure 1.** Distribution of (**A**) degree, (**B**) normalised betweenness, and (**C**) eigenvector centrality values based on cross-sectional household survey data on livestock movement. Only a few households showed high centrality values, highlighting their importance in the network.

The online version of this article includes the following figure supplement(s) for figure 1:

**Figure supplement 1.** A topological network depicting the overall connectivity among study households based on shared resource areas in The Gambia.

**Figure supplement 2.** Spearman's rank correlation analysis illustrating the relationship between household (HH) Rift Valley fever virus (RVFV) seropositivity in cattle and unweighted, normalized (**A**) degree, (**B**) betweenness, and (**C**) eigenvector centrality (computed only on the giant component) values derived from this study.

**Figure supplement 3.** Spearman's rank correlation analysis illustrating the relationship between the geographic distance separating connected households (HH) and their mean Rift Valley fever virus (RVFV) seropositivity of cattle in this study.

This study aimed to develop and apply an SIR model to simulate RVF dynamics in the cattle population of The Gambia, with a focus on incorporating seasonal transhumant movement as a key mechanism in explaining observed RVF epidemiology. We used household survey data to reconstruct network connections between cattle herds at grazing and watering locations and estimated seasonally dynamic transmission matrices for distinct ecoclimatic regions based on livestock movements. These transmission matrices were integrated into the model to estimate basic reproduction numbers and simulate temporal trends and infection magnitudes that approximate observed RVF dynamics. Using estimates of the force of infection (FOI) in specific ecoclimatic regions, we predicted age-dependent seroprevalence and estimated seropositivity waning rates consistent with observed data. By exploring the behaviour of both deterministic and stochastic versions of the same model, this study sought to provide insight into the intersection of livestock movement and ecological conditions that support RVF dynamics in The Gambia.

## Results
### Household herd connectivity
The network analysis revealed variation in between-herd connectivity through shared water and grazing locations. Centrality measures were disproportionately driven by a smaller subset of transhumant herds with relatively high centrality values, highlighting their prominent role within the network (*Figure 1*). The overall connectivity of the network is presented in *Figure 1—figure supplement 1*. However, these connectivity patterns showed no significant correlation with herd-specific RVFV seropositivity (*Figure 1—figure supplement 2*).

Transhumant movements during the dry season covered considerable distances, with herds directed towards the Gambia river from within the Sahelian eco-region. The pairwise geographic distances between connected herds from different study villages ranged from 3.9 to 81.4 km (median 13.5 km). A weak negative correlation (Spearman's R=–0.073, p<0.05) was observed between the geographic distance of connected herds and their mean RVFV seroprevalence (*Figure 1—figure supplement 3*),

**Table 1.** Mean posterior values and 95% credible intervals (CrI) of the estimated parameters.

| Parameter description | Notation | Unit | Mean value | 95% CrI |
|---|---|---|---|---|
| Per capita birth rate | $b$ | day$^{-1}$ | 0.0022 | 0.0012–0.0033 |
| Per capita natural death rate | $\mu$ | day$^{-1}$ | 0.0017 | 0.0010–0.0023 |
| RVF specific mortality rate | $\gamma$ | day$^{-1}$ | 0.0758 | 0.0358–0.1868 |
| RVF recovery rate | $\delta$ | day$^{-1}$ | 0.1161 | 0.0500–0.1917 |
| Scaling factor | $\psi$ | – | 2.0508 | 0.9224–3.1282 |
| Wet season transmission parameter in the Sahelian eco-region | $\beta_{s,wet}$ | * | $1.4\times10^{-6}$ | $6.0\times10^{-7} - 2.2\times10^{-6}$ |
| Wet season transmission parameter in the Gambia river eco-region | $\beta_{r,wet}$ | * | $3.7\times10^{-6}$ | $1.9\times10^{-6} - 5.3\times10^{-6}$ |
| Dry season transmission parameter in the Gambia river eco-region | $\beta_{r,dry}$ | * | $3.1\times10^{-6}$ | $1.7\times10^{-6} - 4.4\times10^{-6}$ |

*(Infected·susceptible·day)$^{-1}$.

as might be expected from a transmission process influenced by both shared environmental factors and livestock management practices.

## Estimated model parameters

The mean posterior values of the eight estimated parameters and their 95% credible intervals (CrI), derived through the approximate Bayesian framework, are summarised in *Table 1*. These posterior means are used in all subsequent simulations.

## Seasonal and regional RVF dynamics

We simulated the timing and magnitude of RVFV infections across three structured cattle subpopulations, which seasonally occupy two eco-regions with distinct ecological characteristics. At the end of the 20-year simulations, the model remains in a transient phase without fully attaining a quasi-equilibrium state (*Figure 2A*). The results exhibited an oscillatory pattern, with recurrent spikes in the proportion of infectious cattle aligning with the wet season. In the Sahelian eco-region, infections peaked at approximately 5% of the cattle population between the 64th and 111th days of the wet season (September to October) (*Figure 2B*), followed by a secondary phase characterised by sharp decline of infection during the dry season, accompanied by gradual decrease in herd immunity.

While both ecoclimatic regions showed similar temporal RVFV infection dynamics during the wet season, simulations indicated sustained low-level transmission in the Gambia river eco-region throughout the dry season (*Figure 2B*). This continued virus activity suggests a potential 'source-sink' effect, as a small fraction (approximately 1%) of the transhumant subpopulation (*T*) returned to the Sahelian eco-region at the start of the wet season while infectious.

At the end of the deterministic simulation, the maximum number of infectious cattle in the *M*, *L*, and *T* subpopulations during the wet season was approximately 4,562, 2,467, and 443, respectively. During the dry season, the minimum numbers of infectious cattle remaining in the *L* and *T* subpopulations within the Gambia river eco-region were approximately 37 and 4, respectively. By simulating seasonal transmission, and incorporating cattle movement matrices, our deterministic model provided a close approximation to the RVFV infection fluctuations demonstrated by the stochastic model (*Figure 2*).

The stochastic simulation indicated that RVFV infection undergoes periodic demographic fade-outs (local extinctions) in the transhumant (*T*) subpopulation, occurring in 73.8% of realisations (738 of 1000) during the dry season in the Gambia river eco-region (*Figure 3*). Each of these realisations predicted between one and five distinct local extinction events (*Figure 3—figure supplement 1*). Despite these fade-outs, RVFV infection persisted within the broader river eco-region through continued transmission in the river resident (*L*) subpopulation, enabling recurrent re-infection of the *T* subpopulation in subsequent years. Across all 1000 stochastic realisations, a total of 1363 local extinction events were recorded in the *T* subpopulation, with a mean transmission duration of 13.9 years. This corresponded to an estimated 9.8% probability of local extinction per dry season within the

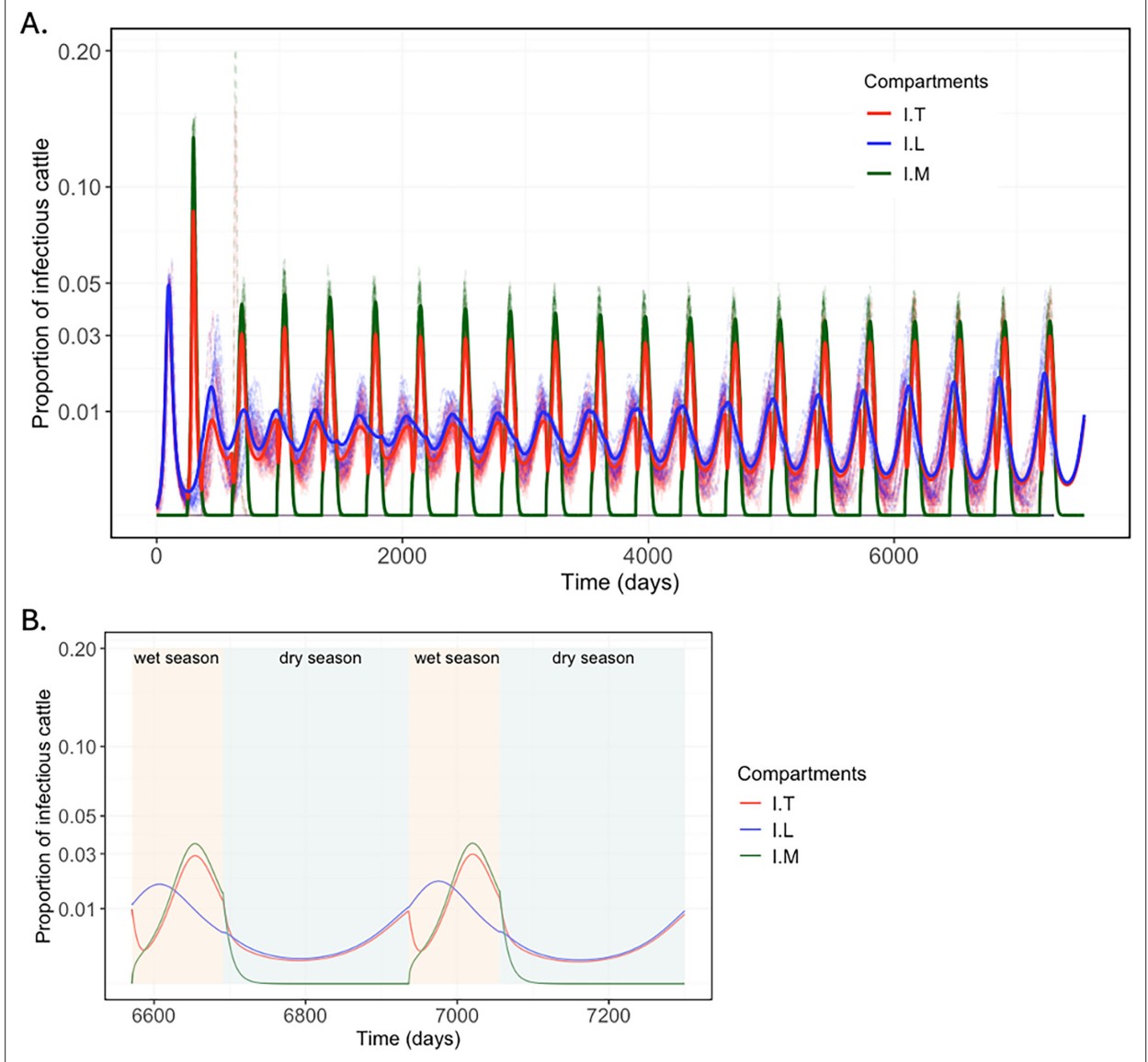

**Figure 2.** Time series of the proportion of infectious cattle in the three subpopulations across the two eco-regions. (**A**) The simulated Rift Valley fever (RVF) infection dynamics from the deterministic model (solid lines) together with 20 realisations of the stochastic model (dashed lines). (**B**) A magnified view highlighting the seasonal dynamics of Rift Valley fever virus (RVFV) transmission focused on the last 2 years of the 20-year deterministic simulation in the cattle subpopulations, highlighting finer seasonal variations (wet season = beige; dry season = cyan). The proportion of infectious cattle peaked at the latter part of the wet season, but infections quickly disappeared in the dry season in the Sahelian eco-region. Once transhumant herds begin to arrive in the Gambia river region, infections are predicted to rise. I=infectious cattle in each subpopulation. A square root transformation was applied to the y-axis for visualisation purposes using coord_trans(y = 'sqrt'), while the data remained in its original scale.

river eco-region. Following these local extinctions, the annual re-introduction of RVFV into the Sahelian eco-region at the start of the wet season, mediated by the *T* subpopulation returning from the Gambia river eco-region, was predicted to occur at a high rate. Finally, complete extinction of RVFV across the entire system occurred in 40.2% of realisations (402 of 1000) before the end of the 20-year simulation period.

Our results showed that, in most realisations, the transhumant subpopulation experienced local extinctions within the first 1000 days, coinciding with the initial transient phase of the RVF dynamics – big outbreaks within a fully susceptible subpopulation. As the system gradually converged towards a quasi-equilibrium state, the probability of RVFV persistence in the *T* subpopulation stabilised (*Figure 3A*). In contrast, RVFV exhibited a higher initial persistence probability within the entire

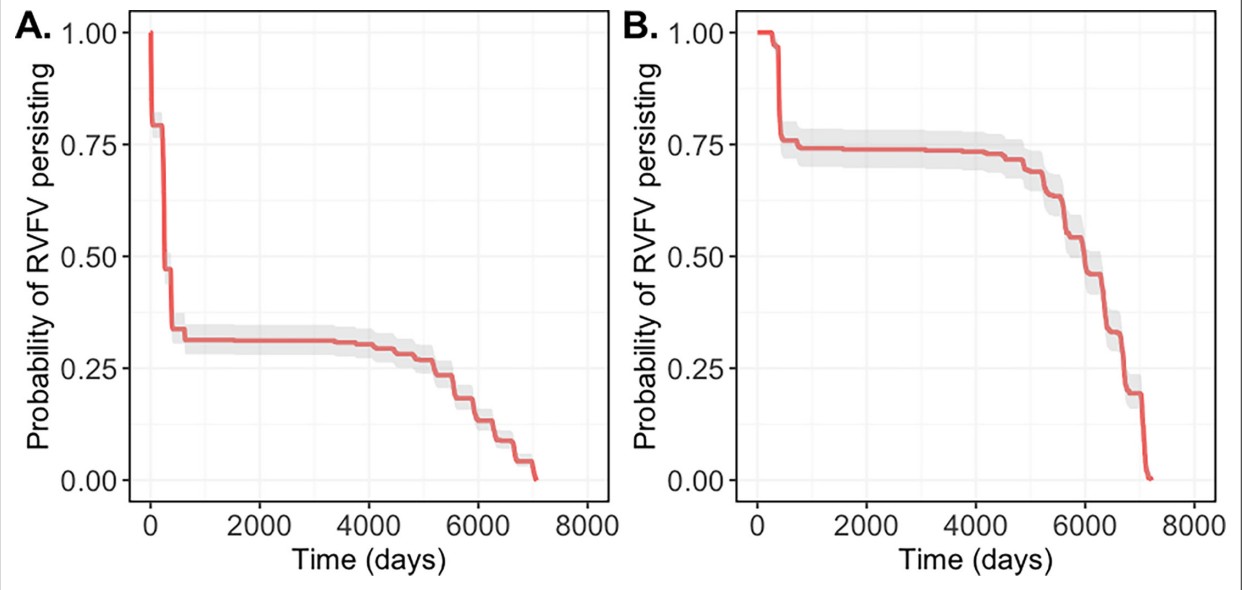

**Figure 3.** Extinction rate of Rift Valley fever virus (RVFV) over time (red line with 95% CI – grey ribbon), in the transhumant subpopulation (**A**) and the entire cattle population within the system (**B**) based on 1000 stochastic realisations, illustrating differences in the timing of local and system-wide extinctions, respectively. Most of the local extinction occurred shortly after RVFV introduction into a fully susceptible population in the *T* subpopulation. Note: the timing of the local extinction in the *T* subpopulation depicted here represents the first extinction event within this subpopulation; re-infection occurs when the subpopulation returned to the river.

The online version of this article includes the following figure supplement(s) for figure 3:

**Figure supplement 1.** Distribution of the number of local extinction events in the *T* subpopulation over an average transmission period of 13.7 years, based on stochastic simulations.

system, with most system-wide extinctions occurring beyond the 5000th day of the simulation as infection dynamics neared a quasi-equilibrium state (*Figure 3B*).

## Estimated $R_{st}$ and $\lambda_{i,season}$

The $R_0$ computed for the entire system assuming a fully susceptible cattle population was 3.15 during the wet season and 1.33 during the dry season. The seasonality of transmission was further illustrated by variations in the $R_{st}$ value (*Figure 4*), which revealed peaks ($R_{st} > 1$) during the wet season across the entire system, aligning with the amplification of infection as epidemiological conditions are assumed to be favourable for transmission. In the latter part of the dry season, $R_{st} > 1$ values were also

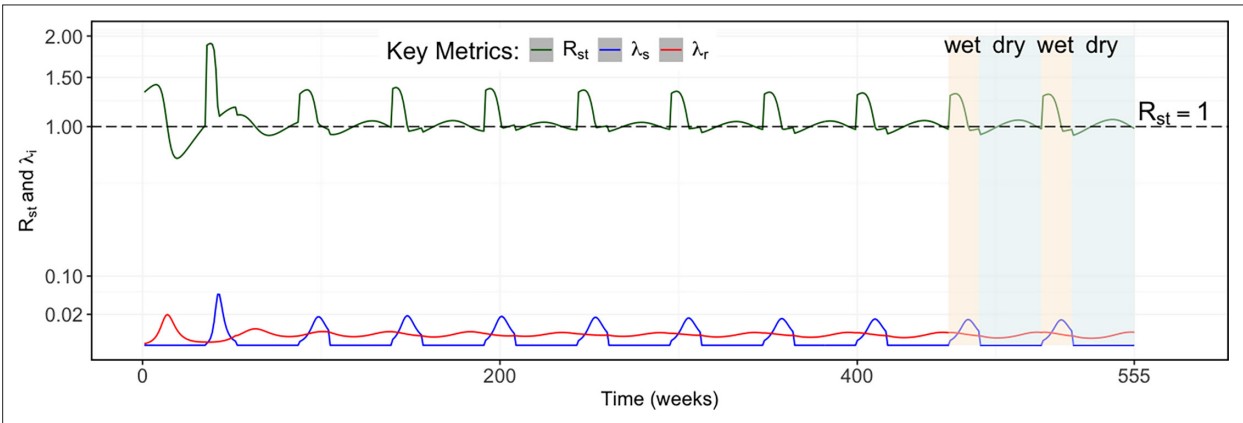

**Figure 4.** The full 10-year simulation of the weekly $R_{st}$ (green) and force of infection $\left(\lambda_{i,season}\right)$ in each eco-region (blue = Sahelian, red = Gambia river eco-region). Shaded areas correspond to the seasons (wet season = beige; dry season = cyan) in the last 2 years of the simulation. A square root transformation was applied to the *y*-axis for visualisation purposes using coord_trans(y = 'sqrt'), while the data remained in its original scale.

**Table 2.** Summary of estimated seasonal reproduction number ($R_{st}$) and regio-specific force of infection $\lambda_i$.

| Seasons | Mean $R_{st}$ | Maximum $R_{st}$ | Minimum $R_{st}$ | Avg. weeks $R_{st} > 1$ |
|---------|---------------|------------------|------------------|--------------------------|
| Wet | 1.15 | 1.40 | 0.91 | 9/17 |
| Dry | 1.02 | 1.11 | 0.87 | 23/35 |

| Seasons | Region | Mean $\lambda_i$ | Maximum $\lambda_i$ | Minimum $\lambda_i$ |
|---------|--------|------------------|---------------------|---------------------|
| Wet | Sahelian | 0.008 | 0.05 | 0.0001 |
|  | river | 0.003 | 0.005 | 0.0002 |
| Dry | Sahelian | – | – | – |
|  | river | 0.002 | 0.018 | 0.0001 |

observed as the susceptible population recovered, supporting the localised persistence of transmission in the Gambia river eco-region (*Figure 4*).

Table 2 summarises $R_{st}$ and $\lambda_{i,season}$ values across both eco-regions and seasons, providing insights into the RVFV dynamics across temporal and spatial scales.

## Predicted age-seroprevalence curve

Here, we fitted a model using the weekly $\lambda_{i,season}$ values estimated for each ecoclimatic region at conditions of quasi-equilibrium to explore the observed trend in age-seroprevalence. Age was used as a proxy for cumulative exposure duration, consistent with prior methodologies (*Salje et al., 2016*). A cohort of newly born calves was subjected to the specific $\lambda_{i,season}$ associated with the eco-region they occupy across their lifespan to estimate the cumulative exposure to RVFV in the absence of waning seropositivity. In the absence of any waning of seropositivity, the model substantially overestimated seroprevalence, predicting values of approximately *M*: 63%, *L*: 64%, and *T*: 84% after 10 years (*Figure 5A*). These predicted seroprevalence values were then re-estimated accounting for potential waning of seropositivity in recovered cattle, enabling reconstruction of the age-stratified seroprevalence profile observed in The Gambia (*Figure 5B*). The posterior distribution of the rate of decay of RVFV seropositivity ($\pi$) yielded a mean value of 0.00217 per week (95% CrI: 0.00216–0.00219). This decay rate corresponded to a seropositivity half-life of approximately 319 weeks (~6 years). These results suggested that the observed seroprevalence was influenced by gradual loss of seropositivity over time in immune cattle.

## Elasticity analysis

A 1% change in the mean posterior value of each of the eight parameters resulted in a corresponding percentage change in predicted seroprevalence of each subpopulation (*Table 3*). The elasticity analysis indicated that the per-capita birth (*b*) and natural death ($\mu$) rates exerted the strongest influence on predicted seroprevalence (*Table 3*). In contrast, parameters related to transmission ($\beta_{s,wet}$, $\beta_{r,wet}$, $\beta_{r,dry}$) and infection dynamics ($\tau$, $\gamma$, and $\delta$) exhibited relatively lower elasticities. Additionally, comparisons between general linear model (GLM) and Loess fits suggested that the relationship between the percentage change in the mean posterior parameter values and the corresponding percentage change in predicted seroprevalence was well approximated as linear (*Figure 6*).

## Discussion

Previous modelling studies have significantly advanced our understanding of RVF epidemiology, particularly regarding the disease dynamics in ruminant hosts and mosquito vectors. These models have informed the development of predictive tools, surveillance frameworks, control interventions, as well as the evaluation of their impact (*Anyamba et al., 2006*; *Gachohi et al., 2016*; *Gaff et al., 2007*; *Tennant et al., 2021*). Despite these advancements, the persistence of RVFV in the Sahelian zone, where unfavourable dry season conditions make permanent RVFV circulation unlikely, remains poorly understood (*Durand et al., 2020*; *Lumley et al., 2017*). Increasing evidence suggests that livestock movements play a critical role in influencing RVF epidemiology in this zone (*Belkhiria et al.,*

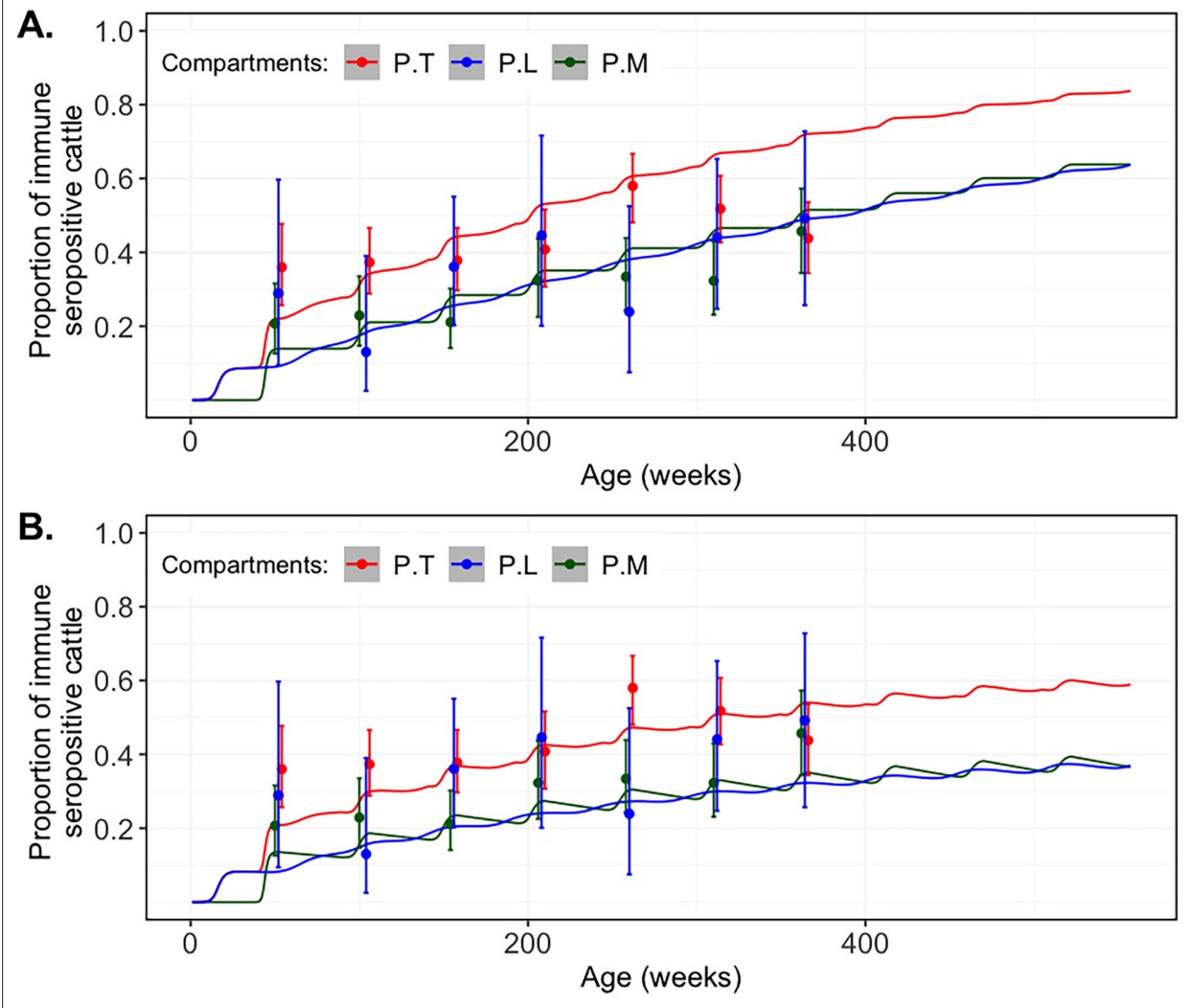

**Figure 5.** Predicted Rift Valley fever virus (RVFV) seroprevalence in each population subject to force of infection (FOI) during the wet and dry seasons. (**A**) Cumulative growth of RVFV immunity over a 10-year period, representing the hypothetical lifespan of cattle, with no consideration for decay of RVFV seropositivity $(\pi = 0)$. (**B**) Proportion of immune cattle after introducing a seropositivity decay parameter $(\pi = 0.00217)$, aligning the predicted seroprevalence with observed data. Observed seroprevalence across different age classes in the three structured cattle populations in The Gambia is shown as dots, with 95% confidence intervals (CI) as error bars. P: cattle exposed to RVFV infection and recovered, assumed to be seropositive.

*2019*; *Jarra et al., 2025*). In this context, risk exposure to RVFV is primarily driven by movements into regions with high environmental suitability for RVFV-competent vectors in search of water and pasture, rather than transmission through direct livestock-to-livestock contact.

To investigate this hypothesis, we developed deterministic and stochastic versions of an SIR model, parameterised using seroprevalence data collected in 2022 in The Gambia, to stimulate RVF dynamics. Our study focuses on the transhumant movement of cattle from the Sahelian ecoclimatic region of the country to the floodplains of the Gambia river as a potential mechanism accounting for the observed RVF dynamics among three cattle subpopulations. Key findings from our study include: (i) seasonal infection patterns peaking at approximately 3% and 1% of cattle in the wet and dry seasons, respectively, in the Gambia river eco-region, whereas infections peaked at approximately 5% in the wet season and disappeared in the dry season in the Sahelian region; (ii) local extinction and re-infection dynamics in the transhumant (*T*) subpopulation, with a high risk (90.2%) of re-introducing RVFV into their homestead villages in the Sahelian eco-region predicted by our stochastic model; (iii) the $R_0$ for the entire system was estimated as 3.15 during the wet season and 1.33 during the dry season, while the mean seasonal reproduction number $(R_{st})$ in both seasons was >1; (iv) the age-seroprevalence

**Table 3.** The elasticity for all eight parameters in the general linear model (GLM).
The parameters with the largest coefficient are shown in bold.

| Parameter | Subpopulation *M* | | Subpopulation *L* | | Subpopulation *T* | |
|---|---|---|---|---|---|---|
| | Intercept | Coefficient | Intercept | Coefficient | Intercept | Coefficient |
| $\beta_{s,wet}$ | –0.004 | 0.140 | –0.005 | –0.027 | 0.008 | –0.032 |
| $\beta_{r,wet}$ | –0.014 | 0.018 | 0.064 | –1.023 | 0.059 | –1.485 |
| $\beta_{r,dry}$ | –0.028 | 0.180 | 0.074 | 1.010 | 0.056 | 1.604 |
| $\mu$ | –0.104 | **–3.965** | 0.151 | **–4.731** | 0.091 | **–4.251** |
| $b$ | –0.154 | **3.603** | 0.108 | **4.784** | 0.059 | **4.230** |
| $\gamma$ | –0.001 | –1.016 | 0.055 | –1.735 | 0.057 | –1.875 |
| $\delta$ | –0.019 | 0.874 | 0.059 | 1.500 | 0.073 | 1.571 |
| $\tau$ | –0.004 | 0.140 | –0.005 | –0.027 | 0.008 | –0.032 |

data in combination with the predicted FOI in each eco-region implied a gradual decay in seropositivity in recovered cattle over time.

## RVFV inter-epidemic transmission dynamics

Our modelling approach incorporates transhumant cattle movement, specific eco-regional transmission rates, and ecological assumptions. Based on predictions that mosquito-borne transmission is responsible for 84% of RVFV infections in cattle (*Nicolas et al., 2014*), we pre-suppose the existence of an active mosquito population in the Gambia river eco-region (*Snow, 1983*) and show that RVFV infections can persist year-round therein. Despite the relatively low proportion of infectious cattle across the three structured subpopulations, RVFV transmission could be sustained in an endemic state over an extended period. Our simulations predicted there is a 90.2% chance the transhumant subpopulation returning to their homestead villages at the start of the wet season would include a small proportion of infectious cattle (approximately 1% of the subpopulation). At the start and end of transhumance, GPS-tracked cattle herds in Cameroon exhibit active movement behaviour at an average daylight speed of 3–4 km/hr (*Motta et al., 2018*), a pattern corroborated by survey data from transhumant households in The Gambia. Given that RVFV incubation and viraemia last approximately 7–8 days, virus re-introduction to the Sahelian eco-region through the returning transhumant herds is plausible, posing a recurrent transmission risk. However, herd immunity among the Sahelian resident cattle population in our simulation prevents annual large-scale outbreaks.

The possibility of RVFV re-introduction between the two eco-regions is consistent with explanations given for the dissemination of the virus during outbreaks within and across countries through movement of potentially viraemic but asymptomatic ruminant livestock (*Abdo-Salem et al., 2011*; *Shoemaker et al., 2002*). In addition, similar patterns have been observed in northern Senegal (*Cecilia et al., 2020*; *Durand et al., 2020*) and Tanzania (*Sumaye et al., 2013*), where movement of infectious livestock between regions with favourable ecological conditions is thought to maintain RVFV endemicity. Our simulation results of the timing and magnitude of infections are supported by evidence showing RVFV amplification during the wet season, with peak circulation occurring at the latter part of the season in Senegal (*Durand et al., 2020*). The $R_{st}$ reflected this, with mean $R_{st}$ values greater than one in the first half of the wet season (9 of 17 weeks) across all eco-regions, and in over two thirds of the dry season (23 of 35 weeks) in the Gambia river eco-region. Earlier studies conducted during an inter-epidemic period in Mayotte estimated seasonal reproduction numbers broadly consistent with those estimated by our study, including *Métras et al., 2017* (maximum $R_{st}$ = 2.19) and *Tennant et al., 2021* (maximum $R_{st}$ = 2.77).

Traditionally, RVFV epidemiology has been characterised as having episodic outbreaks separated by long inter-epidemic periods (*Murithi et al., 2011*; *Sindato et al., 2014*). However, growing evidence across sub-Saharan Africa suggests that RVFV dynamics do not conform to a single archetype but

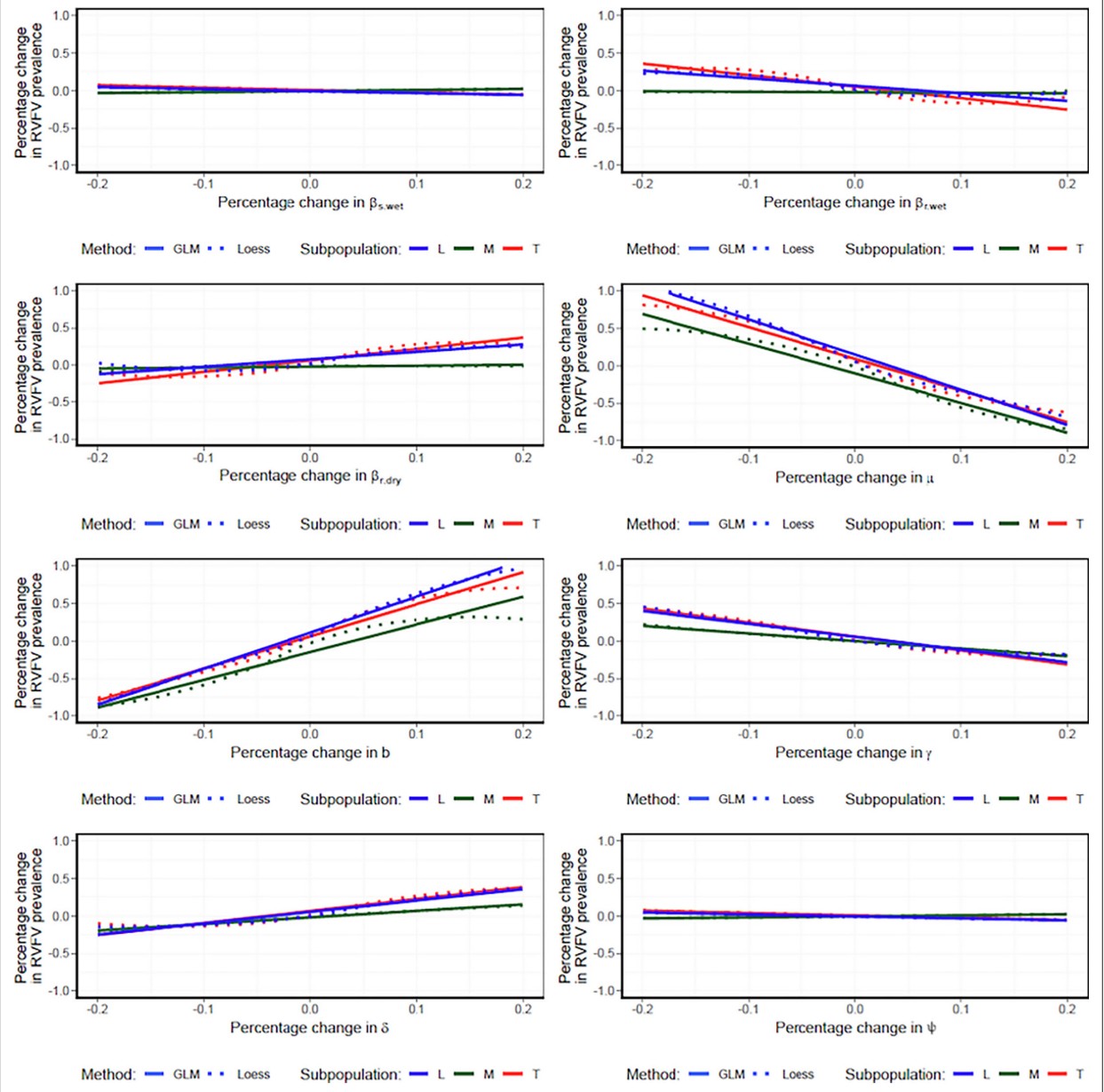

**Figure 6.** The general linear model (GLM) (solid line) and Loess (dotted line) smoothing plots of the relationship between percentage change in predicted seroprevalence of each structured subpopulation (*M*, *L*, and *T*) and percentage change in parameter values constrained within ±20% of the mean posterior value.

instead occur along a continuum (*Rostal et al., 2025*), ranging from epidemic-prone systems where viral persistence is low (*Sindato et al., 2014*) to hyperendemic systems with sustained, moderate transmission without necessarily resulting in substantial outbreaks (*Jarra et al., 2025*; *van den Bergh et al., 2019*). Within this conceptual framework, the stable seasonal dynamics predicted by our model likely position The Gambia towards the hyperendemic end of this spectrum, where ecological conditions, RVFV-competent vectors (potentially including transovarial transmission), and seasonal rainfall maintain transmission. In addition, livestock demography, movement patterns, and acquired immunity may further stabilise these dynamics, thereby supporting long-term circulation without substantial outbreaks.

While our model incorporates the possibility of stochastic extinction, its structure assumes potential for continuous local transmission and therefore tends towards an endemic quasi-equilibrium rather than higher amplitude epidemic cycles. This highlights the plausibility of ecologically sustained

RVFV circulation in cattle in The Gambia with high seroprevalence despite the absence of substantive outbreaks. These findings are consistent with patterns observed in other hyperendemic systems such as northeastern KwaZulu-Natal, South Africa (*van den Bergh et al., 2019*). Nonetheless, longitudinal serological and entomological studies are needed to determine whether the apparent stability reflects true endemic persistence.

## Age-seroprevalence curve and seropositivity decay

Acquired immunity following natural infection with many communicable diseases is often imperfect, with immunity potentially waning over time, offering partial protection or failing entirely (*Le et al., 2021*). Although acquired immunity to RVFV is often considered long-lasting (*Swanepoel and Coetzer., 2004*; *Wilson, 1994*), there is a lack of empirical data regarding the nature and longevity of IgG seropositivity following natural infection in ruminant livestock. In a long-term cohort study, *Wright et al., 2020*, observed that RVFV-specific IgG and neutralising antibodies remained detectable in adult humans for up to 11 years post-infection, although antibody titres progressively declined. This finding has important implications for the interpretation of serological data: while antibodies may be present, the ability to detect them (i.e. seropositivity) may decline over time. Serological diagnostics reliant on detecting specific antibody thresholds may classify immune individuals as seronegative, potentially underestimating the true extent of population immunity. This implication has been observed in other viral infections. For instance, in a 13-year longitudinal study of 34 healthcare workers infected with SARS-CoV, the 50th percentile exhibited anti-nucleocapsid IgG titres below the assay detection limit by year 9, and the 75th percentile fell below the limit by year 13 (*Guo et al., 2020*). These observations reinforce the importance of considering seropositivity decay when interpreting long-term antibody persistence.

Our sero-epidemiological study in The Gambia revealed an age-seroprevalence pattern in cattle that is characterised by high IgG seropositivity during early calfhood, that quickly plateaued or declined with advancing age (*Jarra et al., 2025*). To reconcile the discrepancies between this observed age-seroprevalence pattern and the predictions based on our FOI estimates from each eco-region, we incorporated a seropositivity decay parameter into the model. Specifically, RVFV-exposed cattle in which seropositivity decayed over time transitioned into an immune but seronegative compartment. This adjustment is supported by evidence that cellular immunity (T cell-mediated responses) can confer durable protection even in the presence of substantially low RVFV-specific antibody levels following both natural RVFV infection in humans (*Wright et al., 2020*) and wild-type RVFV challenge in vaccinated mice (*Doyle et al., 2022*). Therefore, our interpretation of antibody decay reflects waning antibody detectability rather than true loss of immunity.

Although earlier RVF seroprevalence-based modelling studies have not explicitly incorporated or contextualised the kinetics of IgG decay, *Le et al., 2021*, provide a valuable theoretical framework of imperfect infection-derived immunity by outlining seven distinct models. These models span between the two classical extremes: the 'perfect' immunity assumed in SEIR frameworks (recovered individuals are permanently protected) and the 'none at all' immunity characteristic of SEIS models (individuals become susceptible again after recovery). Each of these immune decay models produced unique implications for the overall disease dynamics within a population (*Le et al., 2021*). Although a SEIS or SIRS framework could theoretically have allowed recovered cattle to revert to susceptibility, stronger empirical evidence on the duration and durability of natural RVFV antibody responses in ruminant livestock and genuine loss of acquired immune is required to justify this assumption. Future studies incorporating longitudinal serological data could refine parameterisation of our model to explicitly quantify seropositivity decay and its implications for RVFV transmission dynamics, particularly in systems where seropositivity may wane or become undetectable over time despite persistent protection.

## Implications of the model and future enhancements

While this study provides important insight into RVFV dynamics in The Gambia, several data constraints should be considered. The most influential source of uncertainty arises from limited resolution in the household survey data, particularly gaps in spatial data coverage and the coarse temporal information on cattle movements. As a result, the extent of transhumant movements the model relies on may not fully represent the diversity of movements across the country. These limitations may have implications for our conclusions and may lead to under- or over-estimation of periods of heightened

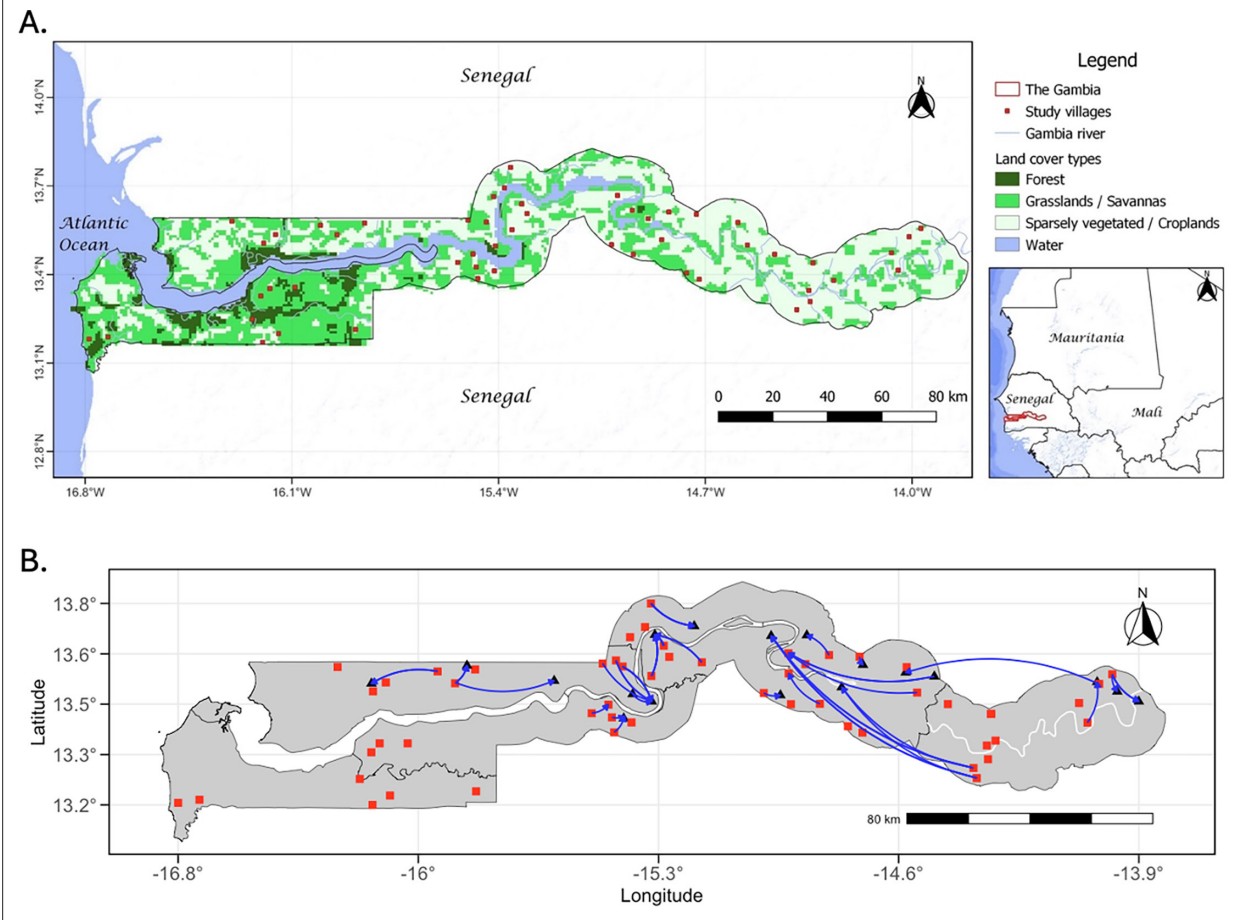

**Figure 7.** Map of study location – The Gambia. (**A**) Illustrates the location of villages selected for household survey and ruminant sampling. Map created in QGIS 3.28.3; land cover data derived from Global Land Cover 2000 (GLC2000) Project. (**B**) Spatial range of transhumant cattle movements in The Gambia identified from the household survey. Movements originated from study villages (represented as red squares) and extended to destination villages, which are either other study villages or villages not selected for this study (represented as black triangles). This study was conducted during the dry season, and all transhumant herds were sampled at their destination villages. Distinct directional movements between homestead villages and their respective destinations are illustrated with blue arrows. The Gambia river is depicted as the meandering white line running the length of the country. The map was generated using the igraph package in R.

transmission risk. Despite these uncertainties, the finding of low-level, spatially diffuse circulation is robust, even though the overall network structure observed suggests a fragmented population with modest but epidemiologically meaningful transhumance distances for The Gambia livestock management systems. Nonetheless, future work incorporating high-resolution movement data, herd-level demographic monitoring, and spatially explicit network models would better capture heterogeneity in contact patterns and improve the accuracy of the predicted RVFV dynamics.

Although RVF can cause substantial abortion and neonatal losses during large outbreaks, our model predicts only low-level endemic circulation in The Gambia. The dampening effect of managed herd demography – through routine sales, replacements, and movement – supports our assumption of near-constant births, consistent with other models of endemic or inter-epidemic RVF dynamics (*Nicolas et al., 2014*). However, this assumption should be revisited in future analyses that incorporate higher-resolution demographic data.

## Methods
### RVFV seroprevalence and household survey data
The data underpinning this modelling study were derived from a previously published cross-sectional serosurvey (*Jarra et al., 2025*), which investigated population-level RVFV seroprevalence patterns

and household-level risk factors associated with seropositivity in ruminant livestock in The Gambia (*Figure 7A*). Briefly, a cross-sectional sero-epidemiological survey was conducted in 2022, collecting 3602 serum samples from ruminant livestock (cattle, sheep, and goats) linked to questionnaire data from 202 livestock-owning households in 52 villages. Details of the study location and data collection procedures are provided in Appendix 1. Due to the very limited number of small ruminants (sheep and goats) involved in transhumant movement in our household survey data, we focus this study on the cattle population. The estimated RVFV-specific IgG seroprevalence in cattle was 36.8% (*n*=1416, 95% confidence interval [CI]: 31.6–40.0), with seroprevalence ranging from 0% to 69.8% (95% CI: 47.1–92.4) in cattle-owning villages (*Jarra et al., 2025*). In the present work, we use this dataset to develop and apply a mechanistic modelling framework aimed at exploring RVFV transmission dynamics across ecological regions.

## Epidemiological setting and cattle population structure

Following *Sumaye et al., 2019*, we define two regions with distinct ecological and climatic characteristics (hereafter referred to as ecoclimatic regions or eco-regions) in The Gambia: the Gambia river ecoclimatic region with its year-round suitability for RVFV transmission contrasts with the Sahelian ecoclimatic region, which experiences seasonal rainfall (July to October) and becomes hot and arid in the dry season (November to June). Our data revealed that 30% of the cattle-owning households participate in transhumant movements from the Sahelian ecoclimatic region to the Gambia river during the preceding dry season (*Figure 7B*). Resident cattle herds are assumed to remain at their homestead villages year-round. This observation led to the classification of cattle into three structured subpopulations: *M*: resident cattle in Sahelian villages, present year-round in this eco-region without access to the Gambia river. *L*: resident cattle in villages within the Gambia river ecoclimatic region, with year-round access to its floodplains. *T*: transhumant cattle in Sahelian villages which seasonally move to the Gambia river eco-region during the dry season and return at the start of the wet season.

The observed RVFV IgG seroprevalence in each of the three populations was estimated as *M*: 30% (95% CI: 27–34), *L*: 36% (95% CI: 28–45), and *T*: 45% (95% CI: 41–48). A 2016 national livestock census of The Gambia estimated the total cattle population at 292,837 (*Department of Livestock Services, 2016*, pers. comms.). The livestock census includes complementary data on livestock management practices and grazing patterns. These data were used to disaggregate the total cattle population across the country into Sahelian resident (*M*), transhumant (*T*), and river resident (*L*) subpopulations, providing the basis for the subpopulation estimates in *Table 4*.

## Movement network analysis

We conducted a descriptive movement network analysis to characterise herd mobility patterns and connectivity within and across eco-climatic regions. The household herd served as the mapping unit to capture both daily local movements around homestead villages and seasonal transhumant migration. Herds within a village typically share grazing and watering resources and were assumed to mix homogeneously with those in neighbouring villages through use of shared resources. Seasonal transhumant

**Table 4.** Preliminary parameter values for the susceptible-infectious-recovered (SIR) model.

| Parameter description | Notation | Value | Unit | Reference |
|---|---|---|---|---|
| Per capita birth rate | $b$ | 0.00215 | day$^{-1}$ | *Gachohi et al., 2016* |
| Per capita natural death rate | $\mu$ | 0.0016 | day$^{-1}$ | *Nicolas et al., 2014* |
| RVF specific mortality rate | $\gamma$ | 0.075 | day$^{-1}$ | *Gachohi et al., 2016* |
| RVF recovery rate | $\delta$ | 1/8 | day$^{-1}$ | *Durand et al., 2020* |
| Scaling factor | $\psi$ | 2 | – | Assumed |
| Decay rate of RVFV seropositivity | $\pi$ | 0.005 | week$^{-1}$ | Assumed |
| Total *M* population | $N_M$ | 177,004 | | 2016 Livestock Census |
| Total *L* population | $N_L$ | 80,433 | | 2016 Livestock Census |
| Total *T* population | $N_T$ | 35,400 | | 2016 Livestock Census |

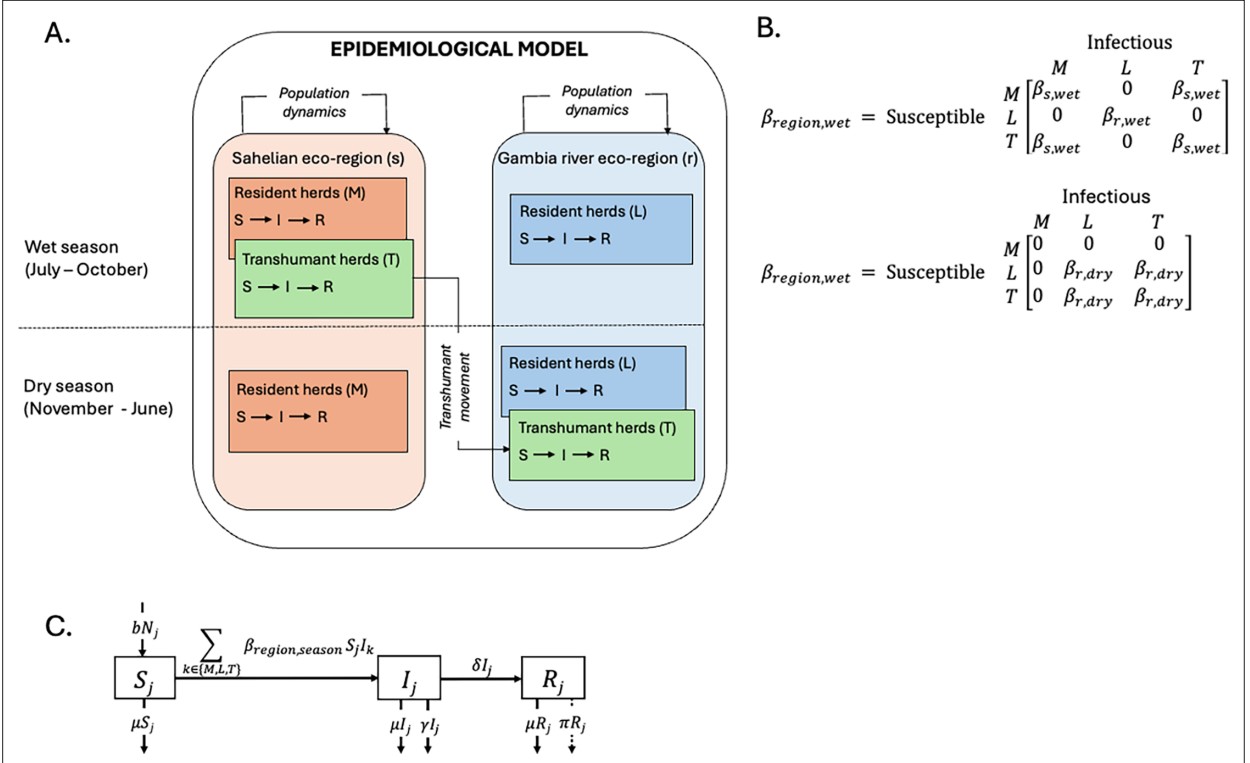

**Figure 8.** Epidemiological model of Rift Valley fever virus (RVFV) transmission and infection dynamics among the cattle subpopulations in The Gambia. (**A**) Schematic representation of ecoclimatic region and seasonal combinations that influence RVFV transmission between the Sahelian areas and the Gambia river eco-region. (**B**) The transmission matrices that determined possible RVFV transmissions between the subpopulations during each season. (**C**) Transmission framework of the within eco-region RVFV transmission; parameters are defined within the main text.

movements established epidemiological links between otherwise distant herds (*Figure 7B*). To represent these movement processes, we constructed an undirected network in which nodes corresponded to herds and edges linked herds that shared grazing or watering locations. For each node, we calculated normalised centrality measures (degree, betweenness, and eigenvector centrality) to quantify relative influence within the network (see Appendix 1). Edge-level geographic distances were estimated using the mean Euclidean distance between connected herds. We then used Spearman's rank correlations to examine associations between network metrics, geographic distance, and herd-specific RVFV seropositivity. This network analysis was used to contextualise movement structure but was not incorporated directly into the transmission model – which was parameterised at the eco-region level to capture broader-scale transmission dynamics.

## General overview of the deterministic model

To better understand and simulate the dynamics of RVFV transmission among the cattle population in The Gambia, we developed a deterministic susceptible-infectious-recovered (SIR) model that incorporated data from the 2022 survey and accounted for transhumant cattle movements across the two different ecoclimatic regions $i$ ($i \in$ {Gambia river eco-region, Sahelian eco-region}). The model operated in continuous time with daily observations and incorporated matrices reflecting cattle movements and region-specific transmission parameters, under the assumption of random mixing among varyingly combined cattle subpopulations within each eco-region (*Figure 8*). The model was required to track the three cattle subpopulations (*M*, *L*, and *T*) as they fuse and fission over the wet and dry seasons in the different eco-regions. The three distinct processes represented in the model were: seasonal cattle movements (arrival and departure of transhumant cattle herds), the population dynamics (birth and death of cattle), and the RVFV dynamics (infection and transmission) in each eco-region (*Figure 8A*).

Entry and exit of cattle through trade or gifting outside the system were not considered. Each structured cattle subpopulation $j$ ($j \in \{M, L, T\}$) was divided into number of susceptible ($S_j$), infectious ($I_j$), and recovered ($R_j$) compartments. The processes driving population dynamics were per-capita birth ($b$) and natural death ($\mu$). Susceptible cattle became exposed to infection at a transmission rate $\beta_{i,season}$ that depends on the eco-region ($i$) and the season (**Figure 8B**). Infectious cattle recovered from RVF at rate $\delta$ or died of the disease at mortality rate $\gamma$ (**Figure 8C**). Birth ($b$) and natural death ($\mu$) were modelled as constant rates to maintain a stable population within each eco-region, reflecting observed herd size in The Gambia over the past 20 years. This approach assumes that RVF does not cause substantial change in cattle numbers under the transmission conditions in The Gambia. In addition, we did not include RVFV-induced abortion in our model. However, the model accounted for temporal variations in population sizes due to transhumant movements and explicitly tracks seasonal population changes in each region:

- **Sahelian eco-region** ($s$): During the wet season, both resident (M) and transhumant (T) cattle are combined $\left(N_{s,wet}\left(t\right) = T + M\right)$. Only resident cattle (M) remain during the dry season $\left(N_{s,dry}\left(t\right) = M\right)$.
- **Gambia river eco-region** ($r$): During the wet season, only the resident cattle ($L$) are present $\left(N_{r,wet}\left(t\right) = L\right)$, while transhumant cattle ($T$) join during the dry season $\left(N_{r,dry}\left(t\right) = L + T\right)$.

We modelled the RVFV transmission rate $\left(\beta_{i,season}\right)$ to account for both mosquito-borne and direct transmission – contact with highly infectious abortion (calving) products. Cattle of all age groups were assumed to participate in the seasonal movement between the two eco-regions, a practice consistently observed among transhumant households over the years. Only movement of the transhumant subpopulation ($T$), serving as the primary epidemiological link between the resident cattle in the Sahelian ($M$) and those in the Gambia river eco-region ($L$), was considered. This necessitated changing the RVFV transmission matrices (**Figure 8B**) between wet and dry seasons to accurately reflect the effects of seasonal movement on RVF transmission dynamics.

## Estimation of model parameters

We estimated basic reproduction numbers ($R_{0,i}$) – the number of secondary infections caused by one infectious cattle in an entirely susceptible population in each ecoclimatic region – based on a standard expression adapted from **Anderson and May, 1991**. This estimation was based on the proportion of the total cattle population in each eco-region that is susceptible ($s_i^*$), assuming homogenous mixing of the cattle subpopulations. Preliminary estimates of seasonal transmission parameters $\beta_{r,wet}$ and $\beta_{r,dry}$ were then calculated as the remaining unknown that can be solved for from a standard expression for $R_{0,i}$ derived from the SIR model (see Appendix 1). RVFV transmission in the West African Sahel is considered to be negligible during the longer dry season (**Chevalier et al., 2004**), and therefore $\beta_{s,dry}$ was set to zero. To align the wet season transmission in the Sahelian eco-region ($\beta_{s,wet}$) with potential year-round transmission in the Gambia river eco-region, a scaling factor $\psi$ was applied to relate the two transmission regimes. $\beta_{s,wet}$ was defined as $\frac{\left(\beta_{r,wet} + \beta_{r,dry}\right)}{\psi}$. This formulation assumes that the annual transmission in the Sahelian eco-region is concentrated into the shorter wet season, whereas transmission in the river eco-region occurs across both wet and dry seasons due to persistent vector habitats. The scaling factor was initially set to 2, due to the absence of an established value in the literature.

Preliminary estimates of four key parameters ($b$, $\mu$, $\delta$, and $\gamma$) were obtained from existing literature (**Table 4**). A final re-estimation of all eight parameters was conducted by fitting the SIR model to the observed seroprevalence using a simulation-based approximate Bayesian computation (ABC) framework. Informative priors for each parameter were defined as triangular distributions, with modes based on the preliminary estimates with upper and lower bounds set to ±70% of these estimates. The rejection method of the ABC algorithm was implemented in the *abc* package in R (version 2.2.1) (**Csilléry et al., 2012**). The priors were used to generate 500,000 randomly parameterised models, and a snapshot RVFV seroprevalence (summary statistics) was predicted for each model at the end of each simulation. These predicted seroprevalence values were then compared to the observed RVFV seroprevalence for each cattle subpopulation. Sets of proposed parameters were accepted for the 0.2% best fitting models (as determined by the Euclidean distance between predicted and observed

seroprevalence values for each subpopulation). The final parameter values were taken as the means of the posterior distributions and used for subsequent model analysis and simulations.

The RVFV dynamics were mathematically formalised using ordinary differential equations (ODEs). A full description of the equations is provided in Appendix 1. An epidemic was initiated at $t$=0 by introducing one infectious cattle into each previously RVFV-free subpopulation. The simulation was run for 20 years (representing the approximate period from the first reported RVFV outbreak in The Gambia in 2002 to our serological survey in 2022). The model incorporated distinct seasonal values of the transmission parameters ($\beta_{i,season}$) (*Figure 8B*). The dry season spanned 245 days (days 1–245), while the wet season lasted 120 days (days 246–365) of each year. The model iteratively checked the time variable daily to apply the appropriate seasonal parameters. The ODEs are implemented and solved in R statistical software (v4.2.2) (*R Development Core Team, 2022*) using the package *deSolve* (*Soetaert, 2010*). All plots are generated using *ggplot2* (*Wickham, 2009*).

## Estimating the per capita rate of decay of RVFV seropositivity

The FOI ($\lambda_{i,season}$) represents the per capita rate at which susceptible cattle become infected with RVFV within each eco-climatic region during the wet and dry seasons. In this study, FOI was estimated mechanistically as a function of the infectious cattle population, rather than being inferred directly from the serological data. A modified version of our transmission model, in which cattle age was substituted for time, was used to generate predicted age-seroprevalence profiles for each subpopulation within each eco-region under quasi-equilibrium conditions (see Appendix 1). Given that cattle typically remain infectious for approximately 1 week, temporal variation in $\lambda_{i,season}$ was evaluated at weekly intervals over a 10-year period, corresponding to the assumed lifespan of cattle in the model. RVF-induced mortality influenced FOI indirectly by reducing the size and duration of the infectious compartment and therefore does not enter explicitly into the FOI formulation.

Cattle that recover from RVFV infection were assigned to an immune seropositive compartment ($P_j$), and the age-seroprevalence curves predicted were compared with observed data. To adjust for inconsistencies between the predicted and observed curves, the predicted age-seroprevalence in each subpopulation was fitted to the observed seroprevalence by including a decay parameter ($\pi$), which described the rate of waning of RVFV seropositivity in recovered cattle over time (*Figure 8C*). The proportion of immune cattle in which seropositivity decayed over time transitioned to an immune seronegative compartment ($D_j$) (see Appendix 1).

The age-seroprevalence model was initialised with a fully susceptible population of newly born calves, ageing over a 10-year lifespan, and subjected to seasonal variation in the $\lambda_{i,season}$ observed at conditions of quasi-equilibrium. A preliminary decay rate of 0.005 per immune cattle per week was set, and a final value was estimated by fitting the predicted seroprevalence to the observed data using ABC. Triangular priors were assigned to $\pi$, with limits extending ±70% from the preliminary rate. The posterior mean of $\pi$ was used to calculate the half-life of RVFV seropositivity decay in cattle in The Gambia.

## Estimation of basic and seasonal reproduction numbers for the entire system

The basic reproduction number ($R_0$) for the fully susceptible cattle population in the entire system during the wet and dry season was calculated by numerically solving for the leading eigenvalues of a matrix **Q**, using the next-generation matrix method described by *Diekmann et al., 2010* (see Appendix 1). Additionally, to account for dynamic susceptibility and transmissibility within the cattle population at close to quasi-equilibrium, a seasonal reproduction number ($R_{st}$) was computed at weekly intervals over the course of the 10-year simulation. At each weekly time step, the model output provided the number of new infected cattle in each partially susceptible subpopulation, which is used to determine $R_{st}$.

## Stochastic modelling and RVFV extinction risk

To complement the deterministic solution, a parallel stochastic simulation was conducted to capture demographic stochasticity arising from the inherent probabilistic dynamics of RVFV transmission. The model employed the tau-leap method (*Gillespie, 2007*), in combination with the Gillespie algorithm to improve computational efficiency. Within our model, the expected rate for each event type (e.g.

demographic processes, transmission events, and infection transitions) was calculated based on the current state of the system, and the number of events occurring within a time step ($\tau = 0.1$ simulated day) was then assumed to be Poisson-distributed. The states of all subpopulations were iteratively updated according to the number of sampled events.

We approximated the extinction risk of RVFV in two contexts: (i) within the transhumant subpopulation ($T$) returning to the Sahelian eco-region at the start of each wet season (stochastic fade-out/local extinction) and (ii) across all three subpopulations (system-wide extinction). For this purpose, 1000 independent stochastic realisations of the model were conducted, to ensure statistical robustness. RVFV extinction risks were quantified by recording the frequencies at which the number of infectious cattle declined to zero within the transhumant subpopulation (see Appendix 1) and across the entire system. It is important to note that the impact of environmental stochasticity was not considered in this analysis.

## Elasticity analyses

We evaluated the sensitivity of the predicted RVFV seroprevalence in each of the three subpopulations to variations in the eight posterior parameter values: $b, \mu, \gamma, \delta, \psi, \beta_{s,wet}, \beta_{r,wet}, \beta_{r,dry}$. Each parameter was independently sampled 1000 times from a uniform distribution, constrained within ±20% of its mean posterior value, while the remaining seven parameters were held at their respective mean posterior values. For each modified parameter set, a new predicted seroprevalence was estimated for each subpopulation. The percentage change in predicted seroprevalence was calculated as the difference between the predicted seroprevalence under the original posterior means and the predicted seroprevalence obtained with the modified parameter value. Likewise, the percentage change in parameter value was calculated as the deviation of the randomly sampled parameter value from its mean posterior value. The elasticity was defined as the percentage change in the seroprevalence (the response variable) per unit percentage change in the parameter value (the explanatory variable) and estimated using a GLM. Additionally, a Loess smoothing function was applied to evaluate the adequacy of the linear approximation and to identify potential nonlinear trends.

## Conclusion

This study contributes to our understanding of RVFV epidemiology in The Gambia by integrating transhumant cattle movement into a modelling framework that accounted for eco-regional transmission dynamics. Our findings highlighted seasonal transmission patterns, with year-round infections predicted in the Gambia river eco-region, reinforcing the role of ecological suitability in sustaining RVFV. While transhumance played a role in shaping RVFV transmission through a high risk of RVFV re-introduction, herd immunity limited large-scale outbreaks in the Sahelian eco-region. The predicted seropositivity decay underscored the importance of considering waning of antibody titres in serological assessments. Future refinements incorporating high-resolution livestock movement data and environmental variability will enhance predictive accuracy. There is a need for strategies informed by transhumance practices in managing RVF, particularly in remote livestock production communities in The Gambia. Understanding livestock owners' perceptions of seasonal movements and the determinants of these movements could inform the design of socially acceptable measures such as targeted vaccination and the strategic or pre-emptive application of topical acaricides for ruminants during peak vector activity in the wet season or before transhumant movements into the river floodplain.

## Acknowledgements

We thank the communities and livestock-owning households for participating in the study. We are also deeply grateful to the livestock assistants of the Department of Livestock Services in The Gambia for their contributions to the field work.

## Additional information

### Funding

| Funder | Grant reference number | Author |
|---|---|---|
| Commonwealth Scholarship Commission | GMCS-2020-721 | Essa Jarra |

The funders had no role in study design, data collection and interpretation, or the decision to submit the work for publication.

### Author contributions

Essa Jarra, Conceptualization, Data curation, Formal analysis, Funding acquisition, Validation, Investigation, Visualization, Methodology, Writing – original draft, Project administration, Writing – review and editing; Divine Ekwem, Conceptualization, Data curation, Formal analysis, Visualization, Writing – review and editing; Sarah Cleaveland, Conceptualization, Supervision, Project administration, Writing – review and editing; Daniel T Haydon, Conceptualization, Formal analysis, Supervision, Methodology, Project administration, Writing – review and editing

### Author ORCIDs

Essa Jarra (ID) https://orcid.org/0000-0002-4358-1218

Reviewer #1 (Public review): https://doi.org/10.7554/eLife.107346.3.sa1

Author response https://doi.org/10.7554/eLife.107346.3.sa2

## Additional files

### Supplementary files

MDAR checklist

### Data availability

The codes that support the findings and produce the figures of this manuscript are available on Github: https://github.com/enz-j/Rvfv_dynamics_gambia (copy archived at *enz-j, 2025*).

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

# Appendix 1

## Rift Valley fever dynamics in a Gambian transhumant cattle system

### Study area

This study was conducted across the six administrative regions of The Gambia, an approximately 11,000 km² semi-enclave of Senegal. As the smallest country in mainland Africa, its dominant geographical feature is the Gambia river, which flows for 450 km across the elongated country supporting a rich ecosystem, including mangrove swamps and wetlands for livestock grazing.

The Gambia has a tropical climate with distinct wet and dry seasons. Due to its location at the edge of the Sahara Desert, rainfall is irregular, occurring in a short unimodal season from July to October, with annual rainfall varying between 650 and 1200 mm (*Touray, 2022*). Wet seasons are occasionally characterised by usually heavy rainfall and high humidity. Conversely, the dry season, from November to June, is hot and arid with average temperatures of around 35°C.

### Data collection

A cross-sectional survey of ruminant livestock linked with household-based questionnaire data was used to gather data on livestock management practices associated with increased odds of RVF seropositivity across The Gambia (*Jarra et al., 2025*). Livestock management follows a traditional mixed farming system combining crop husbandry and livestock rearing. Cattle are grazed unrestrictedly on communal resource areas in the vicinity of villages daily, while small ruminants (goats and sheep) are kept near the compounds. At night, cattle are tethered on the village outskirts, and small ruminants are housed within the compounds.

A total of 52 villages were selected using a multi-stage cluster sampling approach from the list of villages in the 2016 National Livestock Census. Within each selected village, up to four livestock-owning households, serving as the primary units of analysis, were randomly selected for participation using the Livestock Census records. Following informed consent, structured interviews were conducted with household heads to gather detailed information about the seasonal movements of their livestock. A pre-tested, standardised questionnaire was used to ensure consistency across study sites, capturing data on seasonal movement (destinations, durations, and routes) of the household animals during the dry season (November to June). To facilitate effective communication and ensure that participants could express themselves comfortably, discussions were held in the local language of the participant.

### Network analysis

Following are the definitions of parameters used to illustrate household connectivity.

**Degree:** Measures the number of connections to and from each set of nodes (household, and grazing and watering locations) in the 6 months preceding the study. In our network, connections between households and grazing and watering locations were represented by directed edges. Put simply, households with more connections reflected higher levels of interaction at grazing and watering locations within the network.

**Betweenness:** Betweenness centrality reflects the extent to which households within one village occupy strategic positions that influence connection to other villages in the network through the shortest paths. In our study, households with high weighted betweenness played a vital role in epidemiologically linking others in different villages and shaping the network's overall structure. Betweenness scores ranged from 0 (lowest centrality) to 1 (highest centrality).

**Eigenvector centrality:** Indirectly measures centrality based on how connected a node is to other nodes in the network. In our study, we used the eigenvector centrality to describe how connected a household is to other well-connected households – describing the importance of each household in the network. A high eigenvector score means a household is connected to many others that are also highly connected.

## Epidemiological modelling

### Estimation of model parameters

By assuming homogenous mixing of the cattle population and a dynamic process at endemic equilibrium, $R_{0,i}$ can be related to the proportion of the total cattle population in each eco-region that is susceptible, $s_i^*$, as follows, according to *Anderson and May, 1991*:

$$R_{0,i} = \frac{1}{s_i^*} \tag{1}$$

The proportion of susceptible cattle in each eco-region was approximated from the observed seroprevalence of each population, $F_j$, and the total number of cattle within the eco-region, $\widetilde{N}_i$, as follows:

$$s_i^* \sim 1 - \left( \frac{\sum_{k \epsilon \{M,L,H\}} F_k N_k}{\widetilde{N}_i} \right) \tag{2}$$

However, $R_{0,i}$ can also be approximated by:

$$R_{0,i} \sim \frac{\beta_i \widetilde{N}_i}{(\mu + \gamma + \delta)} \tag{3}$$

leaving $\beta_i$ as the remaining unknown.

The transmission rates during the wet and dry season were therefore estimated by:

$$\beta_{s,wet} \sim \psi \left( \frac{R_{0,s} (\mu + \gamma + \delta)}{N_{s,wet}} \right) \tag{4}$$

$$\beta_{r,wet} \sim \frac{R_{0,r} (\mu + \gamma + \delta)}{N_{r,wet}} \tag{5}$$

$$\beta_{r,dry} \sim \frac{R_{0,r} (\mu + \gamma + \delta)}{N_{r,dry}} \tag{6}$$

### Ordinary differential equations

The ODEs that mathematically formalise the number of susceptible, infectious, and recovered cattle of the *T* and *M* populations in the Sahelian eco-region during the wet season are as follows:

$$\frac{dS_T}{dt} = bN_T - \mu S_T - \beta_{s,wet} S_T I_T - \beta_{s,wet} S_T I_M \tag{7}$$

$$\frac{dI_T}{dt} = -(\mu + \gamma + \delta) I_T + \beta_{s,wet} S_T I_T + \beta_{s,wet} S_T I_M \tag{8}$$

$$\frac{dR_H}{dt} = \delta I_T - \mu R_T \tag{9}$$

$$\frac{dS_M}{dt} = bN_M - \mu S_M - \beta_{s,wet} S_M I_M - \beta_{s,wet} S_M I_T \tag{10}$$

$$\frac{dI_M}{dt} = -(\mu + \gamma + \delta) I_M + \beta_{s,wet} S_M I_M + \beta_{s,wet} S_M I_T \tag{11}$$

$$\frac{dR_M}{dt} = \delta I_H - \mu R_M \tag{12}$$

The state of the $L$ population in the Gambia river eco-region during the wet season after population and infection dynamics for cattle in the susceptible, infectious, and recovered compartments was modelled by:

$$\frac{dS_L}{dt} = bN_L - \mu S_L - \beta_{r,wet} S_L I_L \tag{13}$$

$$\frac{dI_L}{dt} = -\left(\mu + \gamma + \delta\right) I_L + \beta_{r,wet} S_L I_L \tag{14}$$

$$\frac{dR_L}{dt} = \delta I_L - \mu R_L \tag{15}$$

The ODEs for the susceptible, infectious, and recovered cattle of the $T$ and $L$ populations in the Gambia river valley during the dry season are as follows:

$$\frac{dS_T}{dt} = bN_T - \mu S_T - \beta_{r,dry} S_T I_T - \beta_{r,dry} S_T I_L \tag{16}$$

$$\frac{dI_T}{dt} = -\left(\mu + \gamma + \delta\right) I_T + \beta_{r,dry} S_T I_T + \beta_{r,dry} S_T I_L \tag{17}$$

$$\frac{dR_T}{dt} = \delta I_T - \mu R_T \tag{18}$$

$$\frac{dS_L}{dt} = bN_L - \mu S_L - \beta_{r,dry} S_L I_L - \beta_{r,dry} S_L I_T \tag{19}$$

$$\frac{dI_L}{dt} = -\left(\mu + \gamma + \delta\right) I_L + \beta_{r,dry} S_L I_L + \beta_{r,dry} S_L I_T \tag{20}$$

$$\frac{dR_L}{dt} = \delta I_L - \mu R_L \tag{21}$$

The population dynamics in the $M$ populations in the Sahelian eco-region during the dry season are modelled as follows:

$$\frac{dS_M}{dt} = bN_M - \mu S_M - \beta_{s,dry} S_M I_M \tag{22}$$

$$\frac{dI_M}{dt} = -\left(\mu + \gamma + \delta\right) I_M + \beta_{s,dry} S_M I_M \tag{23}$$

$$\frac{dR_M}{dt} = \delta I_M - \mu R_M \tag{24}$$

## Estimation of region-specific RVFV FOI

We determined the variation in $\lambda_{i,season}$ when the RVFV dynamics is at quasi-equilibrium as:

$$\lambda_{s,wet} = \beta_{s,wet} I_{s,M}\left(t\right) + \beta_{s,wet} I_{s,T}\left(t\right) \tag{25}$$

$$\lambda_{r,wet} = \beta_{r,wet} I_{r,L}\left(t\right) \tag{26}$$

$$\lambda_{r,dry} = \beta_{r,dry} I_{r,L}\left(t\right) + \beta_{r,dry} I_{r,T}\left(t\right) \tag{27}$$

## Estimation of the basic and seasonal reproduction numbers

The framework outlined below for estimating $R_{st}$ also applies for the estimation of $R_0$, with the only difference being the substitution of the number of susceptible cattle in each population at the end of each week, $S_j\left(t\right)$, with the fully susceptible cattle population at the start of the simulation, $N_j$.

$R_{st}$ was estimated as the spectral radius of the matrix **Q**:

$$\mathbf{Q} = -\mathbf{F}\mathbf{T}^{-1} \tag{28}$$

$$\boldsymbol{F_{region,wet}} = \begin{bmatrix} \beta_{s,wet}\,S_M\,(t) & 0 & \beta_{s,wet}\,S_M\,(t) \\ 0 & \beta_{r,wet}\,S_L\,(t) & 0 \\ \beta_{s,wet}\,S_T\,(t) & 0 & \beta_{s,wet}\,S_T\,(t) \end{bmatrix}$$

$$\boldsymbol{F_{region,dry}} = \begin{bmatrix} 0 & 0 & 0 \\ 0 & \beta_{r,dry}\,S_L\,(t) & \beta_{r,dry}\,S_L\,(t) \\ 0 & \beta_{r,dry}\,S_T\,(t) & \beta_{r,dry}\,S_T\,(t) \end{bmatrix}$$

$$\boldsymbol{T} = \begin{bmatrix} -(\mu + \gamma + \delta) & 0 & 0 \\ 0 & -(\mu + \gamma + \delta) & 0 \\ 0 & 0 & -(\mu + \gamma + \delta) \end{bmatrix}$$

$$\boldsymbol{T^{-1}} = \begin{bmatrix} \dfrac{-1}{\mu + \gamma + \delta} & 0 & 0 \\ 0 & \dfrac{-1}{\mu + \gamma + \delta} & 0 \\ 0 & 0 & \dfrac{-1}{\mu + \gamma + \delta} \end{bmatrix}$$

We then numerically solved for $R_{st}$ by calculating the leading eigenvalue of the matrix **Q**. $R_{st}$ values were calculated independently for each season.

## Estimation of the per capita rate of decay of RVFV seropositivity

A version of our SIR model was formulated in terms of cattle age, $a$, to predict RVFV age-seroprevalence profiles for each population subject to the estimated $\lambda_{i,season}$ for the Sahelian and Gambia river eco-regions. The differential equations which governed the seroprevalence dynamics in each population are as follows:

$$\frac{dS_j}{da} = -\lambda_{i,season}S_j \tag{29}$$

$$\frac{dP_j}{da} = \left(\frac{\delta}{\delta + \gamma}\right)\lambda_{i,season}S_j - \pi P_j \tag{30}$$

$$\frac{dD_j}{da} = \pi P_j \tag{31}$$

Note: $\lambda_{i,season}$ signifies the FOI in each eco-region during the wet or dry season.

