## [Editor Report · eLife Assessment]

This modelling study tests several hypotheses describing how seasonality and migration drive the epidemiology of Rift Valley Fever Virus among transhumant cattle in The Gambia. The work is methodologically **solid**, and the findings offer **valuable** insights into how the movement of cattle in and out of the Gambia River and Sahel ecoregions could lead to source-sink transmission dynamics among cattle subpopulations, sustaining endemic transmission.

---

## [Referee Report · Reviewer #1 (Public review)]

Summary:

This study uses data from a recent RVFV serosurvey among transhumant cattle in The Gambia to inform the development of an RVFV transmission model. The model incorporates several hypotheses that capture the seasonal nature of both vector-borne RVFV transmission and cattle migration. These natural phenomena are driven by contrasting wet and dry seasons in The Gambia's two main ecoregions and are purported to drive cyclical source-sink transmission dynamics. Although the Sahel is hypothesized to be unsuitable for year-long RVFV transmission, findings suggest that cattle returning from the Gambia River to the Sahel at the beginning of the wet season could drive repeated RVFV introductions and ensuing seasonal outbreaks. Upon review, the authors have removed an additional analysis evaluating the potential impacts of cattle movement bans on transmission dynamics, which was poorly supported by the methodological approach.

Strengths:

Like most infectious diseases in animal systems in low- and middle-income countries, the transmission dynamics of RVFV in cattle in The Gambia are poorly understood. This study harnesses important data on RVFV seroepidemiology to develop and parameterize a novel transmission model, providing plausible estimates of several epidemiological parameters and transmission dynamic patterns.

This study is well written and easy to follow.

The authors consider both deterministic and stochastic formulations of their model, demonstrating potential impacts of random events (e.g. extinctions) and providing confidence regarding model robustness.

The authors use well-established Bayesian estimation techniques for model fitting and confront their transmission model with a seroepidemiological model to assess model fit.

Elasticity analyses help to understand the relative importance of competing demographic and epidemiological drivers of transmission in this system.

Weaknesses:

The model does not include an impact of infection on cattle birth rates, but the authors justify that this parameter should have limited impact on dynamics given predicted low-level circulation patterns, as opposed to explosive outbreaks, in this region.

The importance of the LVFV positivity decay rate is highlighted but loss of immunity is not considered in the SIR model. The authors do discuss uncertainty regarding model structure and a need for future data collection to begin to answer this question.

The model's structure, including homogenous mixing within each ecoregion and step-change seasonality, allows for estimation of generalized transmission rates at a macro scale. However, it greatly simplifies the movement process itself and assumes that transhumant cattle movement is the only mechanism for RVF reintroduction into the Sahel region. The authors discuss that integration of more finely-scaled movement and contact data may help to address this limitation in future work.

This model seems well-suited to be exploited in future work to explore for e.g. impacts of cattle vaccination, and potential differential efficiency when targeting T herds relative to M or L.

Comments on revisions:

I thank the authors for thoughtfully and thoroughly addressing my concerns. I have no further comments.

---

## [Author Response]

The following is the authors’ response to the original reviews.

**Joint Public reviews:**
(1) Stable annual dynamics vs. episodic outbreaks

We agree that RVF is classically described as producing periodic epidemics interspersed with long inter-epidemic periods, often linked to extreme rainfall events. Our model predicts more regular seasonal dynamics, which reflects the endemic transmission patterns we have observed in The Gambia through serological surveys. In this revision, we have:

- clarified that while epidemics occur in other parts of sub-Saharan Africa, our results are consistent with the epidemiological narrative of RVF in The Gambia, characterised by sustained, moderate transmission without resulting in substantial outbreaks (hyperendemicity).

- discussed how model assumptions (e.g. seasonality, homogenous mixing) may bias our results toward an endemic quasi-equilibrium dynamic.

- highlighted the implications of this for interpretation and for public health decision-making.

(2) Use of network analysis

We acknowledge the reviewer’s concern. The network analysis was conducted descriptively to characterize cattle movement patterns and the structure of herd connections, but it was not formally incorporated into the model. In this revision we have:

- clarified this distinction in the manuscript to avoid overinterpretation.

- emphasized the need for future modelling work using finer-scale movement data, which could support more realistic herd metapopulation dynamics and better capture heterogeneity in transmission.

(3) RVFV reproductive impacts

While RVF outbreaks are known to cause substantial abortions and neonatal deaths, these events occur during sporadic epidemics. In the Gambian context, where we’re not observing large outbreaks but rather low-level circulation, the annual impact of RVF infection on births is likely modest compared to baseline herd turnover. Moreover, cattle demography is partly managed, with replacement and movement buffering birth rates against short-term losses.

Our model includes birth as a constant demographic process, it’s reasonable to assume stable population since we are not explicitly modelling outbreak-scale reproductive losses. This approach is consistent with other RVF transmission models that adopt a similar simplifying assumption. However, we have acknowledged this simplification as a limitation in the revised manuscript.

(4) Missing ODEs for M herds in the dry season

We thank the reviewer for identifying this omission. The ODEs for the M subpopulation in the dry season were not included in the appendix due to an oversight, though demographic turnover was implemented in the model code. We have now added the missing equations to the appendix.

(5) Role of immunity loss and model structure (SIR vs. SIRS)

We acknowledge that the decline of detectable antibodies over time (seropositivity decay) is an important consideration in RVFV serology; however, whether this decline reflects a true loss of protective immunity following natural infection remains unknown. Available evidence suggests that infected cattle likely develop long-lasting immunity, and findings in humans further support this assumption, although longitudinal field data regarding RVFV-specific antibody durability in animals are not available to the best of our knowledge. From a modelling perspective, our objective was to estimate FOI and use it to predict an age-seroprevalence curve consistent with the observed cross-sectional age-seroprevalence patterns. We therefore adopted a parsimonious SIR framework, interpreting loss of seropositivity as a potential explanation for discrepancies between observed and predicted age-seroprevalence rather than explicitly modelling waning immunity. We have now:

- clarified this rationale, emphasising that there is no direct evidence for waning immunity following natural RVFV infection in cattle, although evidence of seropositivity decay has been suggested in human.

- highlighted that while an SEIS/SIRS framework could theoretically generate different long-term dynamics, evaluating this approach requires stronger evidence for true immunity loss.

(6) RVFV induced mortality in serocatalytic model

We thank the reviewer for this comment and for raising an important conceptual point. However, the force of infection in our study is not estimated using a serocatalytic framework. Instead, FOI is estimated mechanistically within the transmission model as a function of the number of infectious cattle, rather than from age-stratified seroprevalence data.

RVF-induced mortality is accounted for through its effect on the infectious compartment, where increased mortality reduces the number and duration of infectious cattle and therefore indirectly reduces FOI. Consequently, RVF-related cattle death does not need to be explicitly incorporated into the FOI expression itself. Seroreversion similarly does not influence FOI estimation under this modelling framework. We have clarified this distinction in the Methods section to avoid confusion between mechanistic transmission models and serocatalytic approaches.

(7) Clarifying previous vs. current study components

We have revised the Methods and Appendix to make clearer distinctions between our previous work (e.g. household survey data collection, seroprevalence estimates) and the analyses undertaken for this manuscript (e.g. model development and fitting).

(8) Limitations paragraph

We have expanded the limitations section to identify the sparse household movement data as contributing most to uncertainty. We have outlined how these limitations may have implications for our conclusions, and may lead to under- or over-estimation of periods of heightened transmission risk.

(9) Movement ban simulations & suitability of model for vaccination interventions

We appreciate the reviewer’s concerns regarding the movement ban simulation. On reassessment, we agree that our model structure might not ideally be suited to exploring a movement ban. In this revised manuscript, we have removed this analysis. We are currently developing separate work focused on RVF vaccination strategies in cattle, where this model structure might be more directly applicable, and will reserve a deeper investigation of vaccination interventions for that forthcoming publication.

**Reviewer #1 (Recommendations for the authors):**

We thank the reviewer for the recommendations regarding the Introduction, Methods, Results, and Supplementary Figures. We have addressed these points below and revised the manuscript accordingly.

(1) Introduction: Should avoid describing as "inaccessible" the regions that are inhabited by nomadic and transhumant pastoralists.

We have revised the wording to “hard-to-reach” regions.

(2) Methods: Can the authors state what share of the animals included in the household survey data were cattle as opposed to other small ruminants? It would be helpful to understand what share of the data is "excluded"

We have now included the total number of cattle sampled, providing clarity on the proportion of data used in the analyses.

(3) Methods: When introducing the deterministic model, it seems unnecessary to mention the initialization conditions (i.e., introduction of a single infected individual at time 0) when this is later repeated in the Estimation of model parameters section, where it seems simulations were first conducted.

We have removed the redundant description.

(4) Results: Could the negative correlation between geographic distance of connected herds and mean seroprevalence simply indicate proximal exposure rather than common risk factors?

We acknowledge that both mechanisms are plausible. RVFV transmission is strongly influenced by share environmental factors that shape mosquito dynamics; however, direct transmission between proximal cattle herds may also occur through close contact with infectious tissues, bodily fluids, or contaminated materials. We have clarified this interpretation in the Results section.

(5) Figure S5: inconsistent notation for the scaling factor parameter (tau), which is expressed in equations and tables as psi.

We thank the reviewer for identifying this issue and have corrected all instances to ensure consistent use of tau throughout the manuscript.

(6) Figure S6: Why a density plot, isn't the number of temporary extinctions (x-axis) discrete?

We have replaced the density plot with a bar plot in Figure S6.